# New traps for the capture of *Aedes aegypti* (Linnaeus) and *Aedes albopictus* (Skuse) (Diptera: Culicidae) eggs and adults

**Karina Rossi da Silva**[1], **William Ribeiro da Silva**[2,3], **Bianca Piraccini Silva**[1☯], **Adriano Nobre Arcos**[3,4☯], **Francisco Augusto da Silva Ferreira**[2,3☯], **Joelma Soares-da-Silva**[5☯], **Grafe Oliveira Pontes**[3,6☯], **Rosemary Aparecida Roque**[6☯], **Wanderli Pedro Tadei**[2,6☯], **Mário Antonio Navarro-Silva**[7☯], **João Antonio Cyrino Zequi**[1]*

**1** Laboratório de Entomologia Médica, Departamento de Biologia Animal e Vegetal, Universidade Estadual de Londrina (UEL), Programa de Pós-Graduação em Ciências Biológicas, Londrina, Paraná, Brasil, **2** Programa de Pós-Graduação em Ciências Biológicas (Entomologia), Instituto Nacional de Pesquisas da Amazônia (INPA), Manaus, Amazonas, Brasil, **3** Laboratório de Controle Biológico e Biotecnologia da Malária e Dengue, Instituto Nacional de Pesquisas da Amazônia, Manaus, Amazonas, Brasil, **4** Programa de Pós-Graduação em Ecologia e Conservação, Universidade Federal de Mato Grosso do Sul (UFMS), Campo Grande, Mato Grosso do Sul, Brasil, **5** Curso de Ciências Naturais, Campus VII, Universidade Federal do Maranhão (UFMA), Codó, Maranhão, Brasil, **6** Centro de Entomologia, Fundação de Medicina Tropical Doutor Heitor Vieira Dourado (FMT–HVD), Manaus, Amazonas, Brasil, **7** Laboratório de Morfologia e Fisiologia de Culicidae e Chironomidae, Universidade Federal do Paraná (UFPR), Curitiba, Paraná, Brasil

☯ These authors contributed equally to this work.
* zequi@uel.br

**Data Availability Statement:** All relevant data are within the manuscript.

## Abstract

The control of arboviruses carried by *Aedes aegypti* (Linnaeus) and *Aedes albopictus* (Skuse) can be performed with tools that monitor and reduce the circulation of these vectors. Therefore, the efficiency of four types of traps in capturing *A. aegypti* and *A. albopictus* eggs and adults, with the biological product Vectobac WG, was evaluated in the field. For this, 20 traps were installed in two locations, which were in the South (Londrina, Paraná) and North (Manaus, Amazonas) Regions of Brazil, from March to April 2017 and January to February 2018, respectively. The UELtrap-E (standard trap) and UELtrap-EA traps captured *A. aegypti* and *A. albopictus* eggs: 1703/1866 eggs in Londrina, and 10268/2149 eggs in Manaus, respectively, and presented high ovitraps positivity index (OPI) values (averages: 100%/100% in Londrina, and 100%/96% in Manaus, respectively); and high egg density index (EDI) values (averages: 68/75 in Londrina, and 411/89 in Manaus, respectively), so they had statistically superior efficiency to that of the CRtrap-E and CRtrap-EA traps in both regions, that captured less eggs and adults: 96/69 eggs in Londrina, and 1091/510 eggs in Manaus, respectively. Also presented lower OPI values (averages: 28%/4% in Londrina, and 88%/60% in Manaus, respectively); and lower EDI values (averages: 10.5/9 in Londrina, and 47/30 in Manaus, respectively). The capture ratios of *Aedes* adults in the UELtrap-EA and CRtrap-EA traps in Londrina and Manaus were 53.3%/29.5% and 0%/9.8%, respectively. UELtrap-EA can be adopted as efficient tool for *Aedes* monitoring due to their high sensitivity, low cost and ease of use.

**Funding:** The authors thank the FAPEAM (Amazonas State Research Support Foundation) to W.R.S, CNPq (National Council for Scientific and Technological Development) Process 440385/2016-4, which was approved by the MCTIC/FNDCT-CNPq/MEC Call -CAPES/MS-Decit/N° 14/2016 - Prevention and fight against the Zika virus to M. A. Navarro-Silva, and the Coordination for the Improvement of Higher Education Personnel (CAPES) (Financing Code 001) for the financial support to K. R. S. The funders had no role in study design, data collection and analysis, decision to publish, or preparation of the manuscript.

**Competing interests:** The authors have declared that no competing interests exist.

## Author summary

*Aedes aegypti* and *Aedes albopictus* are species of mosquitoes responsible for the transmission of several arboviruses that cause infections worldwide. However, there are still no effective and safe vaccines or medications to prevent or treat arboviruses transmitted by these vectors, except for yellow fever. Moreover, current methodologies for monitoring and controlling *A. aegypti* and *A. albopictus* are not fully effective, as evidenced by the increasing cases of the arbovirus transmitted by these mosquitoes or have incompatible costs with the socioeconomic conditions of a large number of people. Thus, the traps tested in this study can be used as more effective and economical tools for monitoring *A. aegypti* and *A. albopictus*, since they are made with low cost material and they showed high efficiency in the capture of eggs, evidenced by the high values of ovitraps positive index and eggs density index, besides that one of the models captured *Aedes* spp. adults in both regions where they were tested. Therefore, the traps have potential for reducing *Aedes* spp. eggs and adults in the environment and sensibility for determining the local infestation index, which can be reconciled with official government strategies for more accurate vector monitoring actions.

## Introduction

Mosquitoes in the family Culicidae, order Diptera, occur in virtually all regions of the planet. This family is divided into two subfamilies (Anophelinae and Culicinae) in which some species are considered vectors of pathogens of medical importance [1], such as *Aedes* (*Stegomyia*) *aegypti* (Linnaeus, 1762) and *Aedes* (*Stegomyia*) *albopictus* (Skuse, 1894) (Diptera: Culicidae). These species are cosmopolitan and capable of becoming infected with various arboviruses that are responsible for disease and death worldwide [2–5].

Although *A. aegypti* is of African origin, its incidence is currently higher in the Americas, Southeast Asia, and the Western Pacific [4,6,7]. In Brazil, it is the main vector of the four dengue serotypes (DENV-1, DENV-2, DENV-3, DENV-4) and the urban yellow fever virus, which occurs throughout the Brazilian territory [2,3,8]. It also transmits Zika virus (ZIKV) and chikungunya (CHIKV), which are responsible for infections and deaths in over 100 countries [3,9–11].

This species has a home habit, with essentially anthropophilic and synanthropic behavior [2,12–14]. Females prefer artificial containers with standing water for laying, such as tires, disposable cups, potted plants and bottles, especially those of dark colors and with rough surfaces [2,15–17]. In these breeding sites, it is often also possible to find eggs of *A. albopictus*, which originated from Asia, where it is the secondary vector of the dengue virus, which has now spread to Africa, the Americas and Europe [3–5,18].

On the American continent, this species has the potential to carry the same arboviruses as *A. aegypti*, in addition to the ability to carry many other arboviruses in laboratory settings [3–5,19]. Currently, it has adapted to rural, suburban and urban spaces, with a preference for urban spaces with greater vegetation coverage and near native or secondary forests [5,20–22].

Tropical and subtropical countries, such as Brazil, are favorable for the proliferation of vector mosquitoes, given the high temperatures and abundant rainfall. Economic and social factors, such as the lack of basic sanitation and inadequate water supply in the peripheries of large urban centers, also contribute to the availability of mosquito breeding sites and consequently to the spread of viruses [6,19,23,24].

The North Region of Brazil has consistently favorable conditions for *Aedes* spp. proliferation since temperatures remain high throughout the year (annual average of 26˚C), with high precipitation (2000 to 3000 mm annually) [25]. Despite having a mild climate (annual average of approximately 22˚C) and well-defined seasons, South Region of Brazil has a predominance of rains and high temperatures in the summer (average annual rainfall between 1250 and 2000 mm) [25], which combined with local structural conditions favor the proliferation of *Aedes* spp.

Currently, there are still no safe and effective vaccines or medicines to prevent or treat all the arboviruses carried by these vectors, except for yellow fever [26,27]. Thus, measures adopted to control these diseases must consist of actions to reduce vector circulation and, consequently, viral circulation [26,28]. However, the current methodologies for monitoring and controlling *A. aegypti* and *A. albopictus* are not fully effective, as evidenced by the increasing cases of the arbovirus transmitted by these mosquitoes, according to the Brazilian Ministry of Health disclosures [29].

In this sense, the use of traps to capture the eggs of *A. aegypti* and *A. albopictus*, which are called ovitrampas (ovitraps) in Brazil, may be an important strategy for reducing vector circulation. This tool can promote both monitoring of vectors as well as allowing the removal of eggs from the environment, providing indices of indirect mosquito abundance and allowing verification of their spatial and temporal distribution through the number of eggs collected [30–34]. In addition, they have been recommended by the Brazilian Ministry of Health to assist in the surveillance and control of *Aedes* spp. [35].

Ovitraps can be optimized by using entomological glue to capture adults, attractive and larvicidal [30,32,33,36]. The grass infusion *Megathyrsus maximus* Jacq is used as an effective attractant; it acts as a potentiator for the effectiveness of the adult traps and egg traps [32,33,37,38]. *Bacillus thuringiensis israelensis* (Bti) formulations, which is a spore-forming entomopathogen bacterium, are attractive as well as larvicidal because this bacterium synthesizes toxic proteins specific to culicid larvae [39–41].

However, the traps available on the market have incompatible costs with the socioeconomic conditions in Brazil, as they are usually coupled with batteries or motors. Therefore, it is necessary to implement traps that are easy to handle and present low cost to public agencies. From this perspective, this study aimed to evaluate the efficiency of different traps for oviposition and capture of *A. aegypti and A. albopictus* adults in field conditions in South and North Regions of Brazil to validate new tools that can be effective and economical for vector monitoring.

## Methods

### Study area

The study was conducted in localities situated in the states of Paraná and Amazonas, South and North Regions of Brazil. In Paraná, the traps were installed around five buildings located on the *Campus* (74,000 m$^2$) of the Federal Technological University of Paraná (UTFPR), in Londrina city (University Restaurant—23˚ 18'28.51 "S 51˚ 6'56.52" W; Block A—23˚ 18'28.24 "S 51˚ 6'54.04" W; Block B—23˚ 18'27.40 "S 51˚ 6'54.34" W; Block P—23˚ 18'27.21 "S 51˚ 6'50.77 "W; Block K—23˚ 18'26.01" S 51˚ 6'48.77 "W) (Fig 1)

In Amazonas, the traps were distributed at five points located at *Campus* I (255,736.49 m$^2$) of the National Institute for Amazônia Research (INPA), in Manaus city (Point 1–3˚ 5'47 "S 59˚ 59'10" W; Point 2–3˚ 5'43 "S 59˚ 59'11" W; Point 3–3˚ 5'40 "S 59˚ 59'15" W; Point 4–3˚ 5'41 "S 59˚ 59'17 "W; Point 5–3˚ 5'42" S 59˚ 59'15 "W) (Fig 1).

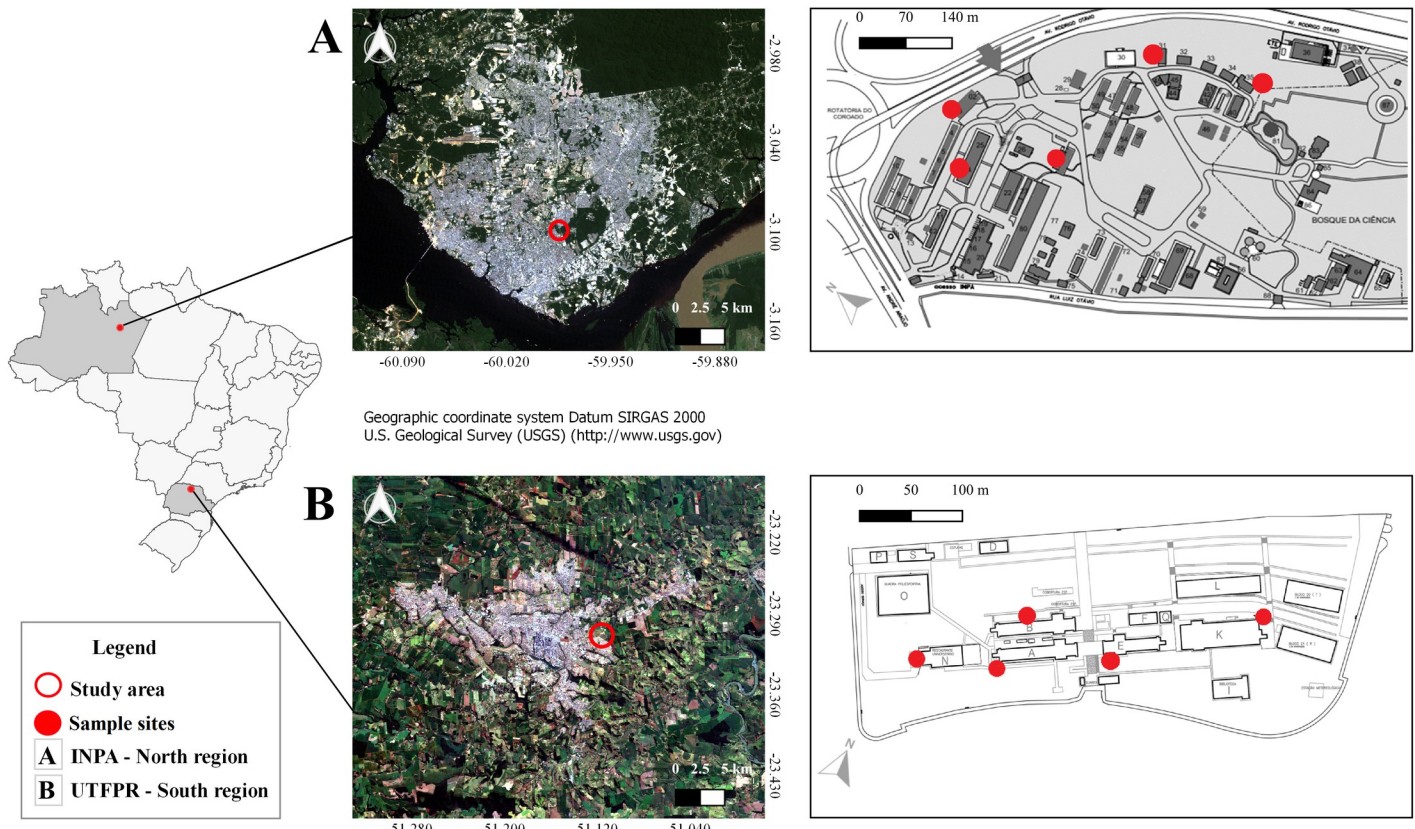

**Fig 1. Location of the study area, demonstrating the distribution of the sample sites in the North and South Regions of Brazil.** Source: https://earthexplorer.usgs. gov/.

The climate of Londrina is humid subtropical, with an average annual temperature of around 22˚C, relative humidity of around 70%, having hot summer and rain in all seasons (annual average between 1400 and 1800 mm) [42]. The study area is situated in the urban perimeter of Londrina and is daily attended by a high number of students and employees, constituting an important area for investigation and monitoring of mosquito vectors of pathogens.

In Manaus, the climate is humid equatorial with an average annual temperature of 26˚C, and a relative humidity of around 80%, with an annual rainfall of around 2,300 mm. The region has two well-defined seasons: rainy (December to June) and dry (July to November), based on rainfall and river levels [43].

The study site is a very wooded urban environment with native forest, which favors *A. albopictus* [2]. As in Londrina, the *Campus* are full of students and employees, being part of the local community.

## Trap characteristics

Four types of traps adapted from the original [44,45] were tested (Fig 2): i) UELtrap-E (standard trap) for egg capture (black rounded plastic vase measuring 12 cm length x 11 cm diameter, with a capacity of 750 mL) (Fig 2A), ii) UELtrap-EA for capture of eggs and adults (12 x 11 cm black rounded plastic vase that is 750 ml in volume, with side openings and a lid with a tulle for ventilation on the top and contains a funnel coated with commercial entomological glue Colly) (Fig 2B), (iii) CRtrap-E for egg capture (clear circular plastic container measuring

**Fig 2.** Traps for capture of *Aedes* eggs and adults under field conditions: A) UELtrap-E; B) UELtrap-EA; C) CRtrap-E; D) CRtrap-EA.

8 cm length x 9 cm diameter, with capacity of 500 mL, contains a black cone with a rough part to facilitate oviposition and egg adhesion) (Fig 2C) and (iv) CRtrap-EA for capturing eggs and adults (8 x 9 cm clear circular plastic container, containing a roughened black outer cone and a lid associated with a funnel coated with commercial entomological glue Colly) (Fig 2D).

The UELtrap-E and UELtrap-EA traps have a Duratree Eucatex reed that measures 13 cm length x 3 cm width, positioned vertically with the rough surface facing upwards to facilitate oviposition and egg adhesion. In contrast, the CRtrap-E and CRtrap-EA have 8 cm length x 1 cm width brown plastic reeds with both smooth and rough surfaces that are placed upright with the rough part facing the outside of the opening from the container.

The traps were optimized for the collection of pregnant females of *Aedes aegypti* and *Aedes albopictus* for the purpose of simplicity and low cost of around US $ 2.5 per unit.

## Collection of eggs and adults of *Aedes* under field conditions

Egg and adult collection at the UTFPR *campus* was carried out for five weeks between March and April (autumn season) of 2017. In contrast, at INPA *Campus* I, the collections were carried out from January to February (rainy season) of 2018 for five weeks. For both Londrina and Manaus, the temperature (˚C), relative humidity (%) and rainfall values (mm) were provide by meteorological database for teaching and research (BDMEP) of the Instituto Nacional de Meteorologia (INMET) [46], whose meteorological station of Londrina (23365116) is located at 11.2 km from the study site (-23.35 S, -51,16 W), while in Manaus, the meteorological station (A101) is located at -3.1 S and -60.02 W.

The sampling design consisted of the installation of four different trap models at each collection point, at ground level in an area that was sheltered from the sun and rain, had little movement of people and animals, and was at a minimum distance of 25 meters from the other traps. Each trap was given 250 mL of water without chlorine and 50 mL of solution containing *M. maximus* (0.11256%) [37] and the biological product Vectobac WG (0.00083%) (Active ingredient: *B. thuringiensis israelensis*), strain AM65-52, 37.4% w/w; Lot No.: 267-853-PG; Date of manufacture: July 2016; Valent BioSciences Corporation—VBC). The attractant solution (50 mL) used in the traps was obtained from a 50 mg/L dilution of the biological product in 5 L of grass infusion (0.0050% and 0.6754%, respectively).

The reeds from UELtrap-E and UELtrap-EA were replaced every seven days and sent in plastic basins containing absorbent paper to the Medical Entomology Laboratory of Londrina State University, and to the Biological Control and Biotechnology of Malaria and Dengue Laboratory at INPA, where the eggs were quantified with the aid of a 50x stereoscope microscope, after drying the reeds at room temperature. The attractive solution of the traps was discarded and replaced every seven days, while the traps were not exchanged.

The eggs present on the plastic reeds and inside CRtrap-E and CRtrap-EA were quantified *in situ* with the aid of a manual magnifying glass (10x) and double-sided tape for egg removal

since the reeds and traps were not replaced. On the other hand, the attractive solution was discarded and replaced every seven days.

Adults collected with UELtrap-EA and CRtrap-EA were removed with the aid of entomological forceps and stored in glass bottles containing absorbent paper to preserve the integrity of the characteristics. Mosquitoes were counted and identified at the species level using external morphological characters with the aid of stereomicroscopy and the identification keys proposed by [1,2,47]. For laboratory and field work, three team members were needed.

Collections in the different study sites were carried out using the Sisbio / Ibama authorization: 23093 and 65287 licenses. In UTFPR Process n˚ 23064.025800 / 2018–21 and in the National Institute of Research of the Amazon the authorization is under number PRJ06.173.

## Data analysis

After quantification of the collected eggs and adults, the OPI–ovitrap positivity index (OPI = N˚ of positive traps/N˚ of examined traps) x 100 [48] and EDI–egg density index (EDI = N˚ of eggs/N˚ of positive traps) [48] were calculated. The data were submitted to the Lilliefors normality test (K samples) and then compared with the data obtained from the evaluated indices (OPI and EDI). Student's t-test (p<0.05) was used for the data with a normal distribution, and the Mann-Whitney test (p<0.05) was used for data that did not present normality. The BioEstat version 5.3 statistical software for Windows [49] was used to assist in all data analysis.

The proportion of female *Aedes* spp. captures in the UELtrap-EA and CRtrap-EA traps were also calculated. This proportion was obtained by calculating the ratio between the total number of eggs and females of *Aedes* spp. collected by the two trap models and considering that each female lays a minimum average of 50 eggs per laying, according to [50] and [51] Thus, let X be the number of females needed to deposit the amount of eggs collected in the traps as follows:

$$X = \frac{number\ of\ eggs\ collected}{minimum\ average\ of\ eggs\ per\ laying} \tag{1}$$

From this, the proportion of female captures of *Aedes* spp. (PC) of the traps is given by the following equation:

$$PC = \frac{number\ of\ females\ caught}{X} . 100 \tag{2}$$

## Results

### Abiotic data recorded in both sampling regions

At the UTFPR *Campus*, the average temperature was 22.6˚C (14.5–30.6˚C), the average relative humidity was 71.5% (44–96%) and the total precipitation was 113.4 mm (0–47.4 mm) throughout the sampling period. At INPA *Campus* I, the sampling period presented an average temperature of 28.3˚C (20.4–35.7˚C), average relative humidity of 80.1% (57.7–95.7%) and total precipitation of 379.6 mm (0–71.3 mm).

### Total eggs and adults of *Aedes* collected at the UTFPR *Campus* in Londrina, Paraná

Considering the traps used exclusively for egg capture, it was observed that in the UELtrap-E traps more eggs were obtained than in the CRtrap-E traps (Fig 3). This result was corroborated when analyzing the average number of eggs obtained for both, since the first obtained an

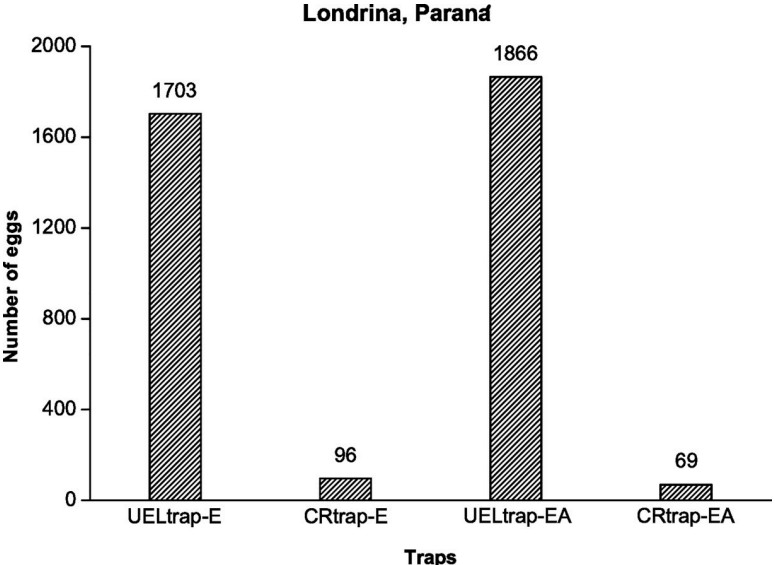

**Fig 3. The total eggs laid by *Aedes* adults in each trap for five weeks from March to April 2017 in Londrina, Paraná, Brazil.**

average (341) 18 times higher than the value (19) obtained in the second trap, thus presenting a statistically significant difference between the respective values (p = 0.0090) (Table 1).

Regarding to the traps that capture eggs and adults, the UELtrap-EA traps presented a higher number of eggs that from the CRtrap-EA traps (Fig 3). This result was also evident because the average egg number (373) obtained by the former was observed to be 27 times higher than the average egg number (14) acquired by the latter; therefore, the differences was statistically significant among the referenced values (p = 0.0107) (Table 1).

Considering the number of eggs of *Aedes* spp. collected in each week of the field experiment, no significant difference (p> 0.05) was found between the average number of eggs acquired in the weeks analyzed in all traps tested (Table 2).

We obtained an OPI value of 100% with the UELtrap-E traps during the collection weeks (Table 3). These results were higher than the values obtained with the CRtrap-E traps during the five-week period. In the latter trap type, the percentage of positivity varied throughout the sampling period, with no eggs in the second week and a higher percentage in the fourth week (0 to 60%). There was a statistically significant difference between the mean OPI values obtained for each trap (p = 0.0090).

**Table 1. Average, maximum and minimum *Aedes* eggs in the different traps from March to April 2017 in Londrina, Paraná, Brazil.**

| Traps | Average (± SD) | Maximum | Minimum |
|---|---|---|---|
| UELtrap-E | 341 (86.9) [A] | 466 | 222 |
| CRtrap-E | 19 (23.7) [B] | 60 | 0 |
| UELtrap-EA | 373 (177.4) [a] | 612 | 179 |
| CRtrap-EA | 14 (12.5) [b] | 33 | 0 |

SD = standard deviation. Different letters in the same column indicate a statistically significant difference (p <0.05) between the average number of eggs obtained for the trap that collects the same stage (eggs or eggs/adults) using Student's t-test or the Mann-Whitney test.

**Table 2. The average and standard deviation of the *Aedes* obtained in each trap from March to April 2017 in Londrina, Paraná, Brazil.**

| Traps | Collection Weeks | | | | |
|---|---|---|---|---|---|
| | 1st | 2nd | 3rd | 4th | 5th |
| UELtrap-E | 44 ± 35 [a] | 69 ± 83 [a] | 93 ± 79 [a] | 70 ± 63 [a] | 65 ± 37 [a] |
| CRtrap-E | 2.4 ± 5 [a] | 0 [a] | 3.4 ± 7,6 [a] | 12 ± 12 [a] | 1.4 ± 1.9 [a] |
| UELtrap-EA | 47 ± 38 [a] | 122 ± 113 [a] | 97 ± 43 [a] | 38 ± 57 [a] | 71 ± 41 [a] |
| CRtrap-EA | 0 [a] | 3 ± 7 [a] | 7 ± 8 [a] | 1 ± 3 [a] | 3 ± 7 [a] |

The same letters on the same row indicate that there was no statistically significant difference (p> 0.05) among the average numbers of eggs obtained each week for all traps tested using Student's t-test or the Mann-Whitney test.

Considering the EDI values obtained per week in the UELtrap-E traps, lower and higher values were found in the first and third weeks (44 and 93), respectively (Table 3). The EDI data obtained by the CRtrap-E traps also varied throughout the sampling period, with the absence of eggs in the second week and a higher quantity in the fourth week (0 and 20). When analyzing the average of the EDI values obtained in UELtrap-E and CRtrap-E, a statistically significant difference (p = 0.0002) was found due to the higher egg density found in the first model (Table 3).

The OPI values obtained from the UELtrap-EA traps were 100% in all weeks analyzed (Table 3). However, for the CRtrap-EA traps, there was variation in the indices, with the absence of eggs in the first week and a higher percentage in the third week (0 and 60%). A significant difference was observed between the mean OPI values between the two trap types tested (p = 0.0009) (Table 3).

Regarding the UELtrap-EA EDI values, the results obtained during the collections showed variations between the indices, with lower and higher values in the fourth and second weeks (36 and 122), respectively (Table 3). The EDI results obtained in the CRtrap-EA traps also showed variations throughout the sampling period, with the absence of eggs in the first week and higher values of eggs in the second and fifth weeks (15), respectively. The egg density in the UELtrap-EA traps was higher than that obtained in the CRtrap-EA traps, which was corroborated by the statistically significant difference (p = 0.0154) (Table 3).

The UELtrap-EA traps captured 17 female specimens; one *A. albopictus*, ten *A. aegypti* and six *Culex quinquefasciatus* Say, 1823. Regarding the percentage of adults collected from each species, 6%, 59% and 35% were found for the species *A. albopictus*, *A. aegypti* and *C. quinquefasciatus*, respectively. According to Eqs 1 and 2, this trap model had a female capture ratio of

**Table 3. Ovitraps positivity index (OPI) and egg density index (EDI) obtained per week in each trap from March to April 2017 in Londrina, Paraná, Brazil.**

| Weeks | UELtrap-E | | CRtrap-E | | UELtrap-EA | | CRtrap-EA | |
|---|---|---|---|---|---|---|---|---|
| | OPI (%) | EDI | OPI (%) | EDI | OPI (%) | EDI | OPI (%) | EDI |
| 1st | 100 | 44 | 20 | 12 | 100 | 47 | 0 | 0 |
| 2nd | 100 | 69 | 0 | 0 | 100 | 122 | 20 | 15 |
| 3rd | 100 | 93 | 20 | 17 | 100 | 97 | 60 | 11 |
| 4th | 100 | 70 | 60 | 20 | 100 | 36 | 20 | 6 |
| 5th | 100 | 65 | 40 | 3.5 | 100 | 71 | 20 | 15 |
| Average | 100[A] | 68[a] | 28[B] | 10.5[b] | 100[A] | 75[a] | 24[B] | 9[b] |

Different letters on the same row indicate a statistically significant difference (p<0.05) between the mean OPI and EDI values obtained for the trap that collects the same stage (eggs or eggs/adults) using Student's t-test or Mann-Whitney test.

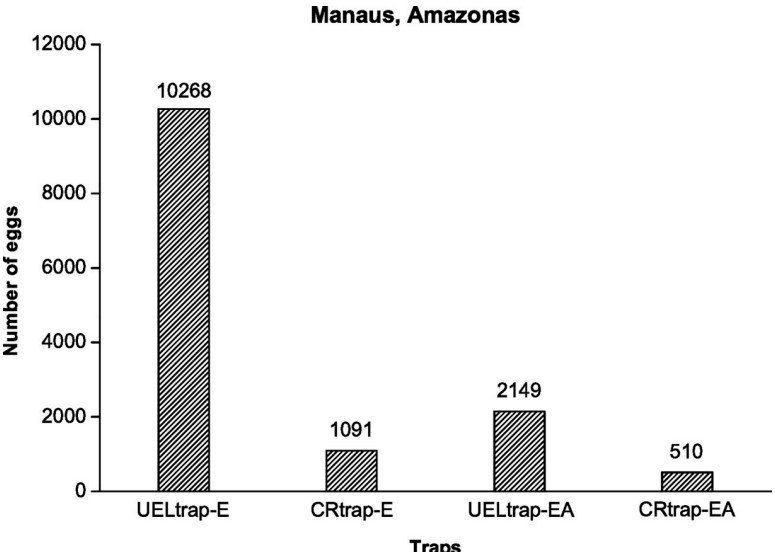

**Fig 4. The total numbers of eggs laid by *Aedes* adults in each trap for five weeks from January to February 2018 in Manaus, Amazonas, Brazil.**

*Aedes* spp. Of 29.50%. This indicated that approximately 29.50% of incoming females were caught. On the other hand, CRtrap-EA captured only one *C. quinquefasciatus* female.

## Total *Aedes* eggs and adults collected at INPA *Campus* I in Manaus, Amazonas

According to the data, UELtrap-E collected more eggs than CRtrap-E (Fig 4). This result was also verified by comparing the averages of the numbers of eggs obtained between the two traps, since the former obtained an average (2054) almost 10 times higher than the value (218) acquired by the latter, with a statistically significant difference (p = 0.0183) between the respective values (Table 4).

Evaluating the quantity of eggs obtained in the traps UELtrap-EA and CRtrap-EA, a higher quantity of eggs was verified in the former trap type (Fig 4). This result was also confirmed by observing that the average number of eggs collected in the former trap type (430) were higher than the average number of eggs verified in the latter trap type (102), which was evidenced by a statistically significant difference between the values (p = 0.0078) (Table 4).

Regarding the average number of eggs obtained by the UELtrap-E traps during each week, a statistically significant difference was found between the values obtained in the first

**Table 4. Average, maximum, and minimum eggs laid by *Aedes* adults in each trap from January to February 2018 in Manaus, Amazonas, Brazil.**

| Traps | Average (± SD) | Maximum | Minimum |
|---|---|---|---|
| UELtrap-E | 2054 (1057) [A] | 3773 | 1068 |
| CRtrap-E | 218 (138) [B] | 363 | 10 |
| UELtrap-EA | 430 (184) [a] | 716 | 221 |
| CRtrap-EA | 102 (82) [b] | 190 | 6 |

SD = standard deviation. Different letters in the same column indicate a statistically significant difference (p <0.05) between the average number of eggs obtained for the trap that collects the same stage (eggs or eggs/adults), Student's t-test or Mann-Whitney test.

**Table 5. The average and standard deviation of the *Aedes* obtained from each trap from January to February 2018 in Manaus, Amazonas, Brazil.**

| Traps | Collection Weeks | | | | |
| --- | --- | --- | --- | --- | --- |
| | 1st | 2nd | 3rd | 4th | 5th |
| UELtrap-E | 755±489[a] | 460±129[a,b] | 310±159[a,b] | 315±132[a,b] | 214±185[b] |
| CRtrap-E | 2±3[a] | 73±71.5[a] | 31±31[a] | 58±58[a] | 55± 55[a] |
| UELtrap-EA | 143±77[a] | 92±121[a,b] | 84±46[b] | 67±51[a,b] | 44±18[b] |
| CRtrap-EA | 1±3[b] | 13±14[a,b] | 37±36[a] | 13±25[a,b] | 38±61[a,b] |

Different letters in the same row indicated a statistically significant difference (p<0.05) between the average number of eggs obtained each week in all traps tested using Student's t-test or Mann-Whitney test.

(755 ± 489) and fifth weeks (214 ± 185) (p = 0.0495). This result was different from that observed in the CRtrap-E trap, where there was no significant difference when comparing the data obtained in each sampling week (p>0.05) (Table 5).

In relation to the average number of eggs obtained in each week of sampling with the use of the UELtrap-EA traps, a statistically significant difference was observed between the values obtained in the first (143 ± 77) and fifth weeks (44 ± 18) (p = 0.0488) as well as between the values obtained for the third (84 ± 46) and fifth weeks (p = 0.0472) (Table 5). Considering the average number of eggs obtained in the CRtrap-EA traps each week, a difference was observed between the first (1 ± 3) and third weeks (37 ± 36) (p = 0.0163) (Table 5).

The OPI values for the UELtrap-E traps demonstrated 100% positive values in all weeks analyzed in the experiment (Table 6). However, in the CRtrap-E traps, the OPI values varied over the sampling period; however, no significant difference was observed between the average OPI values obtained by the two types of traps (p>0.05).

In reference to the UELtrap-E EDI values, there were variations during different sampling weeks, with lower and higher values in the fifth and first weeks (214 and 755), respectively (Table 6). Regarding the EDI values obtained in the CRtrap-E traps, variations were also observed throughout the sampling period, with lower and higher values being observed in the first and second weeks (3 and 73), respectively (Table 6). However, when comparing the average EDI values of the different traps, the results obtained in UELtrap-E were higher than those obtained in CRtrap-E, which was corroborated by a significant difference observed (p = 0.0189).

The OPI values for the UELtrap-EA traps were 100% in four of the five weeks analyzed, except for the fourth week, when this index decreased to 80%. These values are higher than

**Table 6. Ovitraps positivity index (OPI) and egg density index (EDI) obtained per week in each trap from January to February 2018 in Manaus, Amazonas, Brazil.**

| Weeks | UELtrap-E | | CRtrap-E | | UELtrap-EA | | CRtrap-EA | |
| --- | --- | --- | --- | --- | --- | --- | --- | --- |
| | OPI (%) | EDI | OPI (%) | EDI | OPI (%) | EDI | OPI (%) | EDI |
| 1st | 100 | 755 | 60 | 3 | 100 | 143 | 20 | 6 |
| 2nd | 100 | 460 | 100 | 73 | 100 | 92 | 60 | 22 |
| 3rd | 100 | 310 | 100 | 31 | 100 | 84 | 100 | 37 |
| 4th | 100 | 315 | 100 | 58 | 80 | 84 | 60 | 21 |
| 5th | 100 | 214 | 80 | 68 | 100 | 44 | 60 | 63 |
| Average | 100[A] | 411[a] | 88[A] | 47[b] | 96[A] | 89[a] | 60[B] | 30[b] |

Different letters in the same row indicate a statistically significant difference (p<0.05) between the average index (IPO and IDO) obtained for the traps that collect the same stage (eggs or eggs/adults) using Student's t-test or Mann-Whitney test.

 

those obtained in the CRtrap-EA traps, in which varied in each week of collection, with lower and higher values in the first and third weeks (20 and 100%), respectively (Table 6), as evidenced by a significant difference between the average OPI values obtained for the two traps (p = 0.0472).

The EDI values obtained in the UELtrap-EA traps varied during the weeks analyzed in the experiment, showing lower and higher values in the fifth and first weeks (44 and 143), respectively (Table 6). This result was exactly the opposite in the CRtrap-EA traps. Moreover, when comparing the average of the EDI values obtained in each trap, a statistically significant difference was observed (p = 0.0122) due to the higher egg density in the UELtrap-EA traps.

In the UELtrap-EA traps, 25 female specimens were obtained: 23 *A. albopictus*, one *Limatus* spp. and one *Limatus durhamii* Theobald, 1901, representing percentages of 92%, 4% and 4%, respectively. Based on Eqs 1 and 2, these traps presented a female *Aedes* spp. capture ratio of 53.51%. This indicated that approximately 53.51% of the females who entered the traps were caught. On the other hand, in the CRtrap-EA traps, only one *Aedes* spp. female was captured. The capture ratio of *Aedes* spp. female for this trap was 9.80%. Therefore, approximately 9.80% of the females that entered were captured.

## Discussion

When observing the smallest number of eggs and the low EDI and OPI values obtained by the CRtrap-E and CRtrap-EA traps in both study regions, compared to the values obtained by the UELtrap-E and UELtrap-EA traps, it can be seen that the configuration of the first group of traps (smaller blades with less rough surface) may not have provided the ideal conditions for the *Aedes* spp. females to lay eggs.

The light coloration of the traps CRtrap-E and CRtrap-EA may also have influenced egg laying. According to [2,52] females of the genus *Aedes* prefer darker places for oviposition. This fact explains the preference of the females in choosing black traps during egg laying. Therefore, the average numbers of eggs obtained in the UELtrap-E and UELtrap-EA traps and the high values of OPI and EDI observed in both regions indicate that these two trap models (dark color and rough surface) were more inviting to *Aedes* spp. females.

However, when comparing the results obtained for each of the trap models between the two sampling regions, it was evident that all models showed higher efficiency in capturing eggs and adults of *Aedes* spp. in the North Region. This can be explained by the climate of the city of Manaus, where temperatures remain high throughout the year (annual average around 26˚C), in addition to having abundant rainfall, mainly between the months of November and June (rainy season) [25,43,53,54], covering the period in that the collections were carried out in Manaus.

These climate conditions, combined with precarious socio-environmental and economic conditions, frequent in large urban centers like Manaus, provide an ideal environment for the proliferation of mosquitoes, considering the greater availability of breeding sites in these conditions, in addition to the fact that *Aedes* spp. develops faster in a temperature range of 20 to 36˚C, similar to the average in Manaus [6,19,23,53–57].

For the *Aedes* species captured, the high abundance of *A. albopictus* obtained from INPA *Campus* I, Manaus, and the low abundance of this species obtained from the UTFPR *Campus*, Londrina, can be explained by the trap installation environment. In Manaus, the area is composed of fragments of forest reserves suitable for the species, which prefer periurban or urban environments with greater vegetation cover, which is characteristic of wild environments [2,5,21,22]. In a study by [58] in Manaus, a high density of *A. albopictus* was observed in both the central and peripheral regions of the city, where it occurred in areas of urban and

 

periurban forest with anthropogenic alterations and a large number of artificial containers, corroborating the present results. More recently, [33] also observed predominance of *A. albopictus* in a study carried out in the INPA *Campus* I and II.

In contrast, the greater amount of *A. aegypti* caught in Londrina can be explained by the fact that the collection area was more urbanized, unlike the collection area in Manaus, considering that this species is extremely adapted to the urban environment and highly anthropophilic [2,12,14]. These results corroborate with the studies of [59] and [34], which monitored *A. aegypti* in the state of Paraná. In these studies, the authors observed a higher frequency of *A. aegypti* in urban areas, whereas in rural areas, *A. albopictus* was predominant.

The study of [60] also reported that in the municipality of Londrina, Paraná, Brazil, *A. aegypti* populations decreased from urban to rural areas, while the opposite occurred for *A. albopictus*. In a more recent study by [22] in São Paulo, Brazil, there was also a relationship between the occurrence of these species and the type of environment, where the highest density of *A. aegypti* was found in areas with lower vegetation cover, while in areas with higher vegetation cover, *A. albopictus* predominated.

In general, the efficiency of the traps may have been enhanced by the presence of the grass infusion, as it has proven efficacy in attracting *Aedes* spp. compared with the use of only distilled or piped water [32,33,38,61]. The Vectobac WG (*B. thuringiensis israelensis*) biolarvicide used in the experiment as well as other Bti-based products, in conjunction with traps, also can be an important aid for monitoring in view of the proven efficacy of Bti in control of the larvae of *Aedes* spp. Thus, if the larvae hatch from the eggs laid by the females in the reeds, they will not develop into adult form [32,33]. In addition, the effect of Bti comes from four major synergistic toxins (Cry4Aa, Cry4Ba, CryIIAa and CytIAa), which may reduce the likelihood of selection of resistant target organisms [41,62–65], besides not cause damage to other organisms (except Chironomidae and Simuliidae) due to their high specificity for mosquitoes [40,66].

Based on the above, UELtrap-EA has the potential to be used in the monitoring of *A. aegypti* and *A. albopictus* since they were the most collected species and only *Aedes* eggs were collected. This model has high sensitivity for determining the local infestation index and can be implemented in public health programs to reduce both eggs and adults of *Aedes* spp. in the environment along with the UELtrap-E (standard ovitramp), and can be easily transported and used, in addition to having a low cost and high sensitivity for determining the local infestation index.

The results observed for UELtrap-EA in the two study regions also indicated that this trap have efficiency in different environments and seasons, with different climates, demonstrating the possibility for use in different locations and periods of the year. Regarding the CRtrap-E and CRtrap-EA traps, although they presented lower efficiency in capturing the eggs and adults of *Aedes*, they can be optimized by using larger reeds with rougher surfaces for fixing eggs as well as by using darker colors.

These traps do not inconvenience those in the installation areas or to the health workers who should be charged with monitoring the traps since they do not need to be installed indoors but rather in open areas with a large flow of people, such as outside of universities, institutes and industrial buildings as well as in peridomiciles. These traps are an operationally viable and noninvasive method and may become the most effective, practical and economical way to monitor *A. aegypti* and *A. albopictus* on a local scale, provided that the traps are monitored weekly by technical staff.

The entire process can be reconciled with official government strategies for more accurate vector monitoring that can support actions with the population for local surveys and greater efficiency in vector control when necessary.

## Acknowledgments

To the laboratory technicians of the General and Medical Entomology laboratory at the State University of Londrina and the National Research Institute of the Amazon for their assistance in the field.

## Author Contributions

**Conceptualization:** Mário Antonio Navarro-Silva, João Antonio Cyrino Zequi.

**Formal analysis:** Karina Rossi da Silva, William Ribeiro da Silva, Bianca Piraccini Silva, Adriano Nobre Arcos, Francisco Augusto da Silva Ferreira, Joelma Soares-da-Silva, Rosemary Aparecida Roque, João Antonio Cyrino Zequi.

**Funding acquisition:** Wanderli Pedro Tadei, Mário Antonio Navarro-Silva.

**Investigation:** Karina Rossi da Silva, William Ribeiro da Silva, Bianca Piraccini Silva, Adriano Nobre Arcos, Grafe Oliveira Pontes, Wanderli Pedro Tadei, João Antonio Cyrino Zequi.

**Methodology:** Karina Rossi da Silva, William Ribeiro da Silva, Bianca Piraccini Silva, Rosemary Aparecida Roque, João Antonio Cyrino Zequi.

**Resources:** William Ribeiro da Silva, Adriano Nobre Arcos, Francisco Augusto da Silva Ferreira, Joelma Soares-da-Silva, Grafe Oliveira Pontes, Rosemary Aparecida Roque, João Antonio Cyrino Zequi.

**Supervision:** Wanderli Pedro Tadei, Mário Antonio Navarro-Silva.

**Writing – original draft:** Karina Rossi da Silva, William Ribeiro da Silva, Bianca Piraccini Silva, Adriano Nobre Arcos, Francisco Augusto da Silva Ferreira, Joelma Soares-da-Silva, Grafe Oliveira Pontes, Rosemary Aparecida Roque, Mário Antonio Navarro-Silva, João Antonio Cyrino Zequi.

**Writing – review & editing:** Wanderli Pedro Tadei, Mário Antonio Navarro-Silva, João Antonio Cyrino Zequi.

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
