## [Decision Letter · Decision Letter 0]

12 Nov 2020

Dear Dr. zequi,

Thank you very much for submitting your manuscript "New traps for the capture of Aedes aegypti (Linnaeus) and Aedes albopictus (Skuse) (Diptera: Culicidae) eggs and adults" for consideration at PLOS Neglected Tropical Diseases. As with all papers reviewed by the journal, your manuscript was reviewed by members of the editorial board and by several independent reviewers. In light of the reviews (below this email), we would like to invite the resubmission of a significantly-revised version that takes into account the reviewers' comments. 

We cannot make any decision about publication until we have seen the revised manuscript and your response to the reviewers' comments. Your revised manuscript is also likely to be sent to reviewers for further evaluation.

Sincerely,

Claire Donald

Associate Editor

Robert Reiner

Deputy Editor

Reviewer's Responses to Questions

**Key Review Criteria Required for Acceptance?**

**Methods**

-Are the objectives of the study clearly articulated with a clear testable hypothesis stated?

-Is the study design appropriate to address the stated objectives?

-Is the population clearly described and appropriate for the hypothesis being tested?

-Is the sample size sufficient to ensure adequate power to address the hypothesis being tested?

-Were correct statistical analysis used to support conclusions?

-Are there concerns about ethical or regulatory requirements being met?

Reviewer #1: Aedes aegypti and Ae. albopictus are the most well know Culicidae species. However, according to the literature, the difficulties of controlling these species in large and medium-sized cities of Brazil are numerous. Considering their easy proliferation and the limitations to reduce their infestation rates (generated by the complexity of current urban life), to multiply the investigations with this approach is relevant. However, this study has some important issues to be addressed, and the main concerns are its short sampling period and the capture concentration. Thus, I recommend the publication of this study under major revisions.

Reviewer #2: The authors informed that the objectives of the study were "to evaluate the efficiency of different traps for oviposition and capture of A. aegypti and A. albopictus adults in field conditions in South and North Regions of Brazil to validate new tools that can be effective and economical for vector monitoring", but they also concluded that the traps could be useful for reducing eggs and adults of Aedes spp. I agree that the traps could be useful to measure the entomological indicators (and that the authors used suitable methods to evaluate the traps from this point of view), but not as a way to reduce the number of mosquitoes. Furthermore, this was not the objective of the study. For testing this hypothesis, the study would have a different design.

Reviewer #3: The objective of the study is clear and statistical analysis used support the conclusion. However, in the Methods there are some important issues: 

- In study area, the authors need to describe the climate conditional (precipitation, temperature), traps installed environmental conditions (vegetation cover, urban area etc), and others important landscape factors related with Aedes distribution present in the UTFPR and INPA campus. Also, the authors need to include the map with the studies areas and collets points. What were the criteria to choose the buildings to installed the traps? 

-Trap characteristics: Please, include the photos of each traps (UELtrap-E, UELtrap-EA, CRtrap-E and CRtrap-EA) identifying all the elements. Are UELtrap-EA and CRtrap-E designed to collect gravid Aedes female? If yes, the information needs to be in the text. Mention the estimated cost of traps.

- Collection eggs and adults of Aedes under field conditions: Please inform if the traps were and how were cleaned every 7 days? Were attractant solutions replaced every 7 days? Were attractant solutions transported to entomological laboratory to identify immature Aedes (larvae and pupae)? How was the material transported to the laboratory? How many people were needed to monitor the traps (field and laboratory).

Reviewer #4: - The objectives of the study are not clearly articulated with a clear testable hypothesis stated.

- As the traps are adaptations, I missed the specifications and photos to better understand the model, which lack details.

- The sample size was not sufficient to ensure adequate power to address the hypothesis being tested. The installation time of the traps in the field (5 weeks) in a single season does not allow the visualization of their efficacy in different climatic conditions. As a result, important issues of seasonality and trap performance are lost over the course of a year. 

- The study design was not appropriate to address the stated objectives. The traps remained in the same position over the five weeks, which can generate an installation bias. Ideally, they would have passed through a Latin square experimental design. 

- The authors cited only a collection license and did not address ethical issues or even limitations of the study.

**Results**

-Does the analysis presented match the analysis plan?

-Are the results clearly and completely presented?

-Are the figures (Tables, Images) of sufficient quality for clarity?

Reviewer #1: The main concerns are the shortcomings of the study design and limited results. I recommend that the authors make a more accurate description, in addition to referencing an image of the traps, to facilitate the understanding of their functionality. It is important to emphasize that it is not a new trap but characterizes a new adaptation of existing traps, the UELtrap-E and UELtrap-EA, which capture A. aegypti and A. albopictus eggs, respectively.

Reviewer #2: As presented above, the data analysis the authors did was suitable for the evaluation of the traps of the point of view to monitoring the infestation levels of Aedes, but not to conclude that the traps could be useful to control mosquitoes. The authors prioritized, in the Results, the comparison of the traps two by two (UELtrap-E with CRtrap-E and UELtrap-EA with CRtrap-EA). But, taking into account the study design (the four traps put togheter), they colud present the results comparing the four traps. Also, the performance of UELtrap-E was higher than UELtrap-EA in Manaus (a differente result of they found in Londrina), but the authors did not give sufficient attention to this issue in the presentation of results.

Reviewer #3: The results are clear and well organized

with graphics and tables. However there are some important issues:

The proposal of the manuscript is to evaluate the effectiveness of 4 traps for the collection and capture of urban mosquitoes of the genus Aedes. The analyzes were carried out for the aggregated data over time, however the authors could inform the collects points with high and low number of the eggs and adults (male and female) collected per species. What factors could be associated with this finding, even knowing that it is a short period of time (in the discussion).

- Inform how many males and females were caught by adult mosquito species and by trap.

Reviewer #4: - The analysis presented match the analysis plan

- The results clearly and completely presented

- The figures (Tables, Images) are not of sufficient quality for clarity. Photos or schematics were missing for better understanding the traps. Figures 1 and 2 lack subtitles, study period and standard deviation. 

- The first paragraph on the total of eggs and adults collected is actually the discussion of the data and not the presentation of the results.

- CRTrap's performance was better in Manaus than in Londrina, which may be related to the difference in the collection periods and perhaps the installation locations. Since the Latin square model was not considered, one cannot answer about the installation bias.

- In discussion section, line 369 : "The light coloration of the traps CRtrap-E and CRtrap-EA may also have influenced egg laying. According to [2] females ... " they put just the number as a citation, that´s correct? The same im line 400 and other places.

**Conclusions**

-Are the conclusions supported by the data presented?

-Are the limitations of analysis clearly described?

-Do the authors discuss how these data can be helpful to advance our understanding of the topic under study?

-Is public health relevance addressed?

Reviewer #1: The manuscript is relevant to public health. There are limits to the analysis, considering the insufficient quantity of traps used at the different collection points to conduct this research.

Reviewer #2: Taking into account the obtained results, the authors should conclude only about the use of traps to monitor Aedes infestation, but nothing about the use of traps for vector control. The study design used by the authors was not suitable to evaluate this issue.

Reviewer #3: (No Response)

Reviewer #4: - The conclusions was not supported by the data presented. The original ovitrampa also has high sensitivity and specificity and no comparison was made with the standard methodology to compare the results. Thus, the recommendation to use the adapted model as a most effective, practical and economical way to

 monitor A. aegypti and A. albopictus on a local scale is not justified.

- The limitations of analysis was not clearly described

- Public health relevance was addressed

**Editorial and Data Presentation Modifications?**

Reviewer #1: The manuscript "New traps for the capture of Aedes aegypti (Linnaeus) and Aedes albopictus (Skuse) (Diptera: Culicidae) eggs and adults" is relevant to public health and thus should be considered for publication. However, there are issues to be addressed, which I explain below in detail; in this regard, I strongly encourage the authors to make the attached corrections. The main concerns are about the short sampling period and the concentration of captured eggs.

Reviewer #2: I would like to see in the Introduction a more precise justification as to why specifically studying and comparing the traps tested in the study.

It would be good to have maps presenting the trap installation places.

It also would be good for the authors to present figures, photos, or schemes of the four traps evaluated.

Since part of the results in Tables 2 and 5 were also presented in Tables 3 and 6, I ask if Tables 3 and 5 are necessary to be presented.

Reviewer #3: (No Response)

Reviewer #4: (No Response)

**Summary and General Comments**

Reviewer #1: The manuscript "New traps for the capture of Aedes aegypti (Linnaeus) and Aedes albopictus (Skuse) (Diptera: Culicidae) eggs and adults" is relevant and is in line with the scope of the Journal. However, the following modifications need to be addressed to make it publishable:

Comments

Despite the authors' efforts to improve a technology that aims to monitor these species in an alternative and efficient way, I have concerns about it because existing traps perform this same functionality efficiently to achieve these same purposes. For example, concerning the cost of this monitoring tool: is it viable and feasible to manufacture it on a large scale?

Will the authors add the patent registration number of this trap, or is it the object of another patent? It is important to be aware of the legislation of each country.

Abstract Line 33. What were the criteria to define the number of 20 traps to test efficiency?

Line 132-141. These are coordinates of a single point of the area. Please provide the coordinates of at least four points at the boundaries of the studied area, as well as those of the three sampled areas separately.

Lines 142-152. I recommend that the authors make a more accurate description, in addition to referencing an image of the traps, to facilitate the understanding of their functionality. It is important to emphasize that it is not a new trap but characterizes a new adaptation of existing traps, the UELtrap-E and UELtrap-EA, which capture A. aegypti and A. albopictus eggs, respectively.

Line 152. It is not clear which entomological glue is that.

Lines 169-170. What was the criterion used to outline the distances between the collection areas?

Line 248. I believe that these results are precipitated, considering the average time of experimental sampling of the field test. You need to add data from the scientific literature to support this information in a seasonal period for both experimental areas.

Lines 375-381. Could the experimental seasonal period have directly influenced the collection results in both sample areas and thus generated a false, biased efficiency of the traps?

Line 433-439. This is not clear. Why not strengthen the use of traps inside homes, considering that Aedes aegypti is quite frequent inside the houses of urban and suburban areas (where the human population concentration is high)?

Reviewer #2: The methodology used by the authors was suitable to compare the traps, the results they presented corresponded to the objectives of the study and were well discussed. The unique issue that I do not agree with the manuscript is to make a conclusion about the potentiality of the traps to be useful for vector control. The methods used by the authors were not suitable for doing this.

Reviewer #3: (No Response)

Reviewer #4: The proposal to evaluate new tools for Aedes surveillance is important and necessary. However, the design is not adequate to inform the conditions of the study. 

More experiments are needed with a new design. It is suggested to apply the Latin square in different seasons, to better feel the efficacy of the traps. It is important to include ovitraps in the experimental design, as they are the gold standard traps for collecting Aedes eggs.

PLOS authors have the option to publish the peer review history of their article (what does this mean?). If published, this will include your full peer review and any attached files.

Reviewer #1: No

Reviewer #2: No

Reviewer #3: No

Reviewer #4: No
---

## [Decision Letter · Decision Letter 1]

4 Mar 2021

Dear Dr. Zequi,

We are pleased to inform you that your manuscript 'New traps for the capture of Aedes aegypti (Linnaeus) and Aedes albopictus (Skuse) (Diptera: Culicidae) eggs and adults' has been provisionally accepted for publication in PLOS Neglected Tropical Diseases.

Best regards,

Claire Donald

Associate Editor

Robert Reiner

Deputy Editor

Reviewer's Responses to Questions

**Key Review Criteria Required for Acceptance?**

**Methods**

-Are the objectives of the study clearly articulated with a clear testable hypothesis stated?

-Is the study design appropriate to address the stated objectives?

-Is the population clearly described and appropriate for the hypothesis being tested?

-Is the sample size sufficient to ensure adequate power to address the hypothesis being tested?

-Were correct statistical analysis used to support conclusions?

-Are there concerns about ethical or regulatory requirements being met?

Reviewer #1: The main concerns, however, are the short sampling period and the capture concentration.

Reviewer #2: The authors accepted the suggetions I did in the first review of this manuscript. And the answers for this questions are yes.

Reviewer #3: (No Response)

**Results**

-Does the analysis presented match the analysis plan?

-Are the results clearly and completely presented?

-Are the figures (Tables, Images) of sufficient quality for clarity?

Reviewer #1: The result achieved enabled the implementation of an important tool for monitoring Aedes aegypti and Aedes albopictus.

Reviewer #2: The authors accepted the suggetions I did in the first review of this manuscript. And the answers for this questions are yes.

Reviewer #3: (No Response)

**Conclusions**

-Are the conclusions supported by the data presented?

-Are the limitations of analysis clearly described?

-Do the authors discuss how these data can be helpful to advance our understanding of the topic under study?

-Is public health relevance addressed?

Reviewer #1: The conclusions of the work must rely on efficiency in the collection method for monitoring the populations of Aedes aegypti and Aedes albopictus and not on the ability to reduce the circulation of these vectors.

Reviewer #2: The authors accepted the suggetions I did in the first review of this manuscript. And the answers for this questions are yes.

Reviewer #3: (No Response)

**Editorial and Data Presentation Modifications?**

Reviewer #1: (No Response)

Reviewer #2: (No Response)

Reviewer #3: (No Response)

**Summary and General Comments**

Reviewer #1: This is a relevant medical entomology study, based on relavant field data; overall, is well presented.

Reviewer #2: The authors accepted the suggetions I did in the first review of this manuscript.

Reviewer #3: (No Response)

PLOS authors have the option to publish the peer review history of their article (what does this mean?). If published, this will include your full peer review and any attached files.

Reviewer #1: No

Reviewer #2: No

Reviewer #3: No

---

## [Editor Report · Acceptance letter]

9 Apr 2021

Dear Dr. Zequi,

We are delighted to inform you that your manuscript, "New traps for the capture of Aedes aegypti (Linnaeus) and Aedes albopictus (Skuse) (Diptera: Culicidae) eggs and adults," has been formally accepted for publication in PLOS Neglected Tropical Diseases.

Best regards,

Shaden Kamhawi

co-Editor-in-Chief

Paul Brindley

co-Editor-in-Chief
